# SOD2 Enhancement by Long-Term Inhibition of the PI3K Pathway Confers Multi-Drug Resistance and Enhanced Tumor-Initiating Features in Head and Neck Cancer

**DOI:** 10.3390/ijms222011260

**Published:** 2021-10-19

**Authors:** Wei-Ting Hsueh, Shang-Hung Chen, Chia-Hung Chien, Shao-Wen Chou, Pei-I Chi, Jui-Mei Chu, Kwang-Yu Chang

**Affiliations:** 1Department of Oncology, National Cheng Kung University Hospital, College of Medicine, National Cheng Kung University, Tainan 70456, Taiwan; seine@ncku.edu.tw (W.-T.H.); Bryanchen@nhri.edu.tw (S.-H.C.); 2National Institute of Cancer Research, National Health Research Institutes, Tainan 70456, Taiwan; chchien@nhri.edu.tw (C.-H.C.); banacle7187@nhri.edu.tw (S.-W.C.); hunterchi@nhri.edu.tw (P.-I.C.); chujuim@nhri.edu.tw (J.-M.C.); 3School of Medicine, I-Shou University, Kaohsiung 82445, Taiwan

**Keywords:** PI3K/mTOR pathway, target therapy, resistance, ROS, SOD2, tumor-initiating cells

## Abstract

The phosphoinositide-3-kinase (PI3K) pathway has widely been considered as a potential therapeutic target for head and neck cancer (HNC); however, the application of PI3K inhibitors is often overshadowed by the induction of drug resistance with unknown mechanisms. In this study, PII3K inhibitor resistant cancer cells were developed by prolonged culturing of cell lines with BEZ235, a dual PI3K and mammalian target of rapamycin (mTOR) inhibitor. The drug resistant HNC cells showed higher IC_50_ of the proliferation to inhibitors specifically targeting PI3K and/or mTOR, as compared to their parental cells. These cells also showed profound resistance to drugs of other classes. Molecular analysis revealed persistent activation of phosphorylated AKT at threonine 308 in the drug resistant cells and increased expression of markers for tumor-initiating cells. Interestingly, increased intra-cellular ROS levels were observed in the drug resistant cells. Among anti-oxidant molecules, the expression of SOD2 was increased and was associated with the ALDH-positive tumor-initiating cell features. Co-incubation of SOD inhibitors and BEZ235 decreased the stemness feature of the cells in vitro, as shown by results of the spheroid formation assay. In conclusion, dysregulation of SOD2 might contribute to the profound resistance to PI3K inhibitors and the other drugs in HNC cells.

## 1. Introduction

Head and neck cancer (HNC) is a term commonly used to describe different malignant diseases arising from epithelial cells of the upper aerodigestive tract such as the oral cavity and pharyngeal or laryngeal regions. They share common histology features with squamous cell carcinoma, which is the most frequent type. The worldwide incidence of the disease rapidly grew from 2008, with 263,900 cases, to 2018, with 890,000 cases, causing 128,000 and 450,000 deaths, respectively, and ranking sixth as the most common cancer in the world [1,2]. Notably, the age-standardized incidence rate of HNC for males reached 25.7 per 100,000 populations in Taiwan during the years from 2000 to 2012, which was approximately 2.5-fold higher than the world standard in that period [3]. The leading factor in the regionally high incidence is a mixed consequence attributable to prevalent betel quid consumption, which is commonly known to cause oral health problems [4]. The chemical substances within betel quid were shown to be harmful to oral mucosae by eliciting inflammation and cell cycle alteration in normal keratinocyte, eventually propogating tumorigenesis [5]. Indeed, it has been shown that betel nut chewing has a much higher tendency to induce oral leukoplakia and submucosal fibrotic change compared to other well-known carcinogens, such as cigarette smoking and alcohol consumption, in a case-control study [6].

New discoveries in oncology showed certain aberrations in molecular signals that are critical for tumors to survive. Targeting drugs against these factors provide therapeutic opportunities that are more specific to tumors with less interference to the normal tissue. The specific strategies differ from conventional chemotherapies, of which the given doses are often limited by toxicities to reach the optimal treatment dose or must be administered via alternative routes to minimize the adverse effects [7,8,9]. An example of targeting drugs is the inhibitor against epidermal growth factor receptor (EGFR), which is a receptor tyrosine kinase known to elicit intracellular signaling cascades when binding with specific extracellular ligands [10]. A wide range of cancer diseases appear to have EGFR highly expressed, and in some of them, such as HNC and colon cancer, are highly reliant on this signaling pathway for their survival [11]. As a result, a monoclonal antibody cetuximab was developed and approved for the treatment of these diseases [12,13]. However, resistance to targeted therapy is commonly observed and it is known to be in part caused by molecular alterations in cancer cells. For example, altering the expression of different EGFR isoforms or subtypes may impede the anti-cancer effects of the EGFR inhibitors [14,15]. In addition, dysregulated and constitutive activation of an intracellular signaling pathway, caused by mutation of one of the downstream signaling molecules, like *KRAS*, has been shown to cause cetuximab failure in colon cancer [16]. Similarly, upregulation of the PI3K/AKT signaling pathway by aberrant expression of mutated *PIK3CA* genes or by loss of the regulator PTEN was also associated with cetuximab resistance [17]. All these studies have emphasized the importance of routine gene examination for predicting the effect of the anti-EGFR antibody. Notably, the above-described pathways are also widely known for their pathological roles in different HNC diseases [14,18,19,20,21,22]. Thus, compounds targeting molecules within these signaling pathways have been considered as the next-generation drugs for disease control [23,24].

As the PI3K signaling pathway plays multiple roles in the maintenance of cell survival, inhibition of this pathway brings advantages in tumor control [25]. One of the attractive class of anticancer agents is the quinolone derivatives that targets both PI3K and mTOR simultaneously [26]. Of such, BEZ235 has demonstrated certain anticancer effects in a few cancer types. This dual inhibitor was reported to be more potent in cancer cells with PI3K aberrations, such as the H1047R mutation, than to class I PI3K specific inhibitor [27], exhibiting advantage of use since the aberrant PI3K pathway has commonly been shown to be related to the development of the disease [21]. Although recent efforts were made to improve the pharmacological properties of BEZ235 [28], we noticed its complexity in terms of the mechanism can lead to chances for cells to develop cross-resistance as a consequence when the drug fails [29]. The underlying mechanism for resistance to develop is not clear, but it has been suggested that enrichment of cells with stemness features may play a role in the development of drug resistance [30].

In this study, we analyzed the cellular and molecular characteristics, especially the stemness features, of HNC cells with and without PI3K inhibitor resistance. The possible mechanism of drug resistance was studied in these HNC cells. Finally, we proposed a potential treatment strategy to minimize the chance of developing drug resistance, which could warrant future studies.

## 2. Results

### 2.1. Multi-Drug Resistance Is Induced by Prolonged Incubation of BEZ235 in HNC Cells 

Human FaDu and UMSCC1 HNC cells were used in the current study. To develop the resistant variants, cells were cultured with prolonged incubation of BEZ235. As shown in Table 1, the BEZ235-resistant cells showed higher IC_50_ of BEZ235, as compared to their parental cells. In addition, the BEZ235-resistant cells showed profound resistance to LY294002 and rapamycin, which are PI3K and mTOR specific inhibitors, respectively. These cells also exhibited resistance against the EGFR inhibitor, gefitinib. These results confirmed that multi-drug resistant HNC cells were developed by prolonged PI3K/mTOR pathway blockage.

To determine the mechanism of resistance, we investigated the expression of various molecules of the PI3K/mTOR signaling pathway. As shown in Figure 1, expression of the phosphorylated mTOR (s2481) and AKT (t308 and s473) was decreased in FaDu and UMSCC1 cells treated with BEZ235, as expected. Similar results were observed in the FaDu- and UMSCC1-dervided drug resistant cancer cells, except for the expression of the phosphorylated AKT (t308) site, which showed increased expression upon the BEZ235 treatment. These results indicated that incomplete blockage of the PI3K/mTOR signals signaling pathway might play a role in the development of drug resistance.

### 2.2. The Drug Resistant HNC Cells Exhibit Features of Tumor-Initiating Cells

The AKT/mTOR signaling pathway is associated with functioning of the cancer stemness features [31]. Increased expression of ALDH, which could be labeled by ALDEFLUOR, is one of the markers for tumor-initiating cells. Here, the FaDu and UMSCC1-dervided drug-resistant cells both showed increased expression of ALDH (Figure 2A). Results of the Western blot analysis also showed increased expression of Bmi1 and Sox2, which are markers for tumor-initiating cells, in the drug resistant cells (Figure 2B). Interestingly, results of the real-time PCR also revealed increased expression of the Nanog, Oct4, Sox2, and Bmi1 mRNA transcripts (markers of the tumor-initiating cells) in the drug resistant cells (Figure 2C). As tumor-initiating/cancer stem cells are known to exhibit multi-drug resistance [32], our results suggested that the multi-drug resistant HNC cells developed in our study may possibly be the tumor-initiating cells, a subpopulation of cells within the original FaDu/UMSCC1 cell populations.

### 2.3. Increased SOD2 Is Noted in Drug Resistant HNC Cells

Numerous drugs are known to be capable of causing ROS imbalance, which is harmful for cells. Strict regulation of the ROS level is crucial for cells in limiting oxidative damage to maintain their viability, and cells with enhanced ability in dealing with oxidative stress are known to have a survival advantage over those without such ability. Our results showed that BEZ235 increased the levels of ROS in the drug resistant cells (Figure 3A). As SOD2 plays an important role in regulating the levels of ROS (i.e., an anti-oxidant) in cells and its expression is known to be regulated by the PI3K pathway, we postulated that the drug resistance might exhibit an altered regulation on the expression of SOD2 [33]. Here, the drug-resistant cells showed enhanced protein expression of SOD2 (Figure 3B), as compared to the parental drug sensitive cells. Results of the qPCR analysis also revealed an increased mRNA transcript level of SOD2, but not SOD1, which is a cytoplasmic counterpart from the same family, in the FaDu-derived drug-resistant cells as compared to the parental FaDu cells (Figure 3C). Notably by analyzing a publicly available cancer microarray dataset available in an online database (ONCOMINE; www.oncomine.org), we found that higher SOD2 expression was generally predominantly expressed, compared to SOD1 and catalase, in HNC tumors versus in normal tissue (Figure 3D). These data suggested the pivotal role of the anti-oxidant molecule, SOD2, in association with resistant features and the development of HNC disease [34,35,36,37,38,39,40,41,42].

### 2.4. Inhibition of SOD2 Results in Reduced Stemness Features

We next investigated the role of SOD2 in inducing the stemness features in the drug resistant cells. We found that a subgroup of ALDH-positive FaDu cells exhibited higher SOD2 expression (Figure 4A). More importantly, short-term inhibition of the PI3K pathway could induce SOD2 expression (Figure 4B). Because previous reports have shown associations between SOD2 and the cancer stemness features [43,44,45], we postulated the treatment could have adverse effect in bringing cell tolerance, and hence, selecting subgroups of stem-like cells that impede further treatment. We subsequently found that co-incubation of SOD inhibitors with BEZ235 could significantly suppress the development of spheroid cells mass formation (Figure 4C,D). Collectively, results of our study suggested a potential treatment strategy to minimize accumulation of these specific aggressive subsets by combination therapy.

## 3. Discussion

Treatment of advanced HNC is often unsuccessful as affected by the induction of drug resistance. Here, we showed that the drug resistant cells have higher SOD2 expression levels, and PI3K/mTOR inhibition increased the SOD2 levels in cells. Failure to inhibit the pathway completely had profound resistance against the anti-tumor treatments that have similar mechanisms as well as those with different mechanisms (Figure 1), causing the cells to survive multiple drug treatments (Table 1). The consequence was enhancement of cells with tumor-initiating cell properties (Figure 2). The increased cellular level of SOD2 and the higher tolerance to ROS in the resistant cells implied its functional significance (Figure 3A,B and Figure 4A). SOD2 has been frequently reported mandatory for cancer stemness features [43,44,45]. Supportively, inhibition of SOD2 led to cells with the aforementioned features decreased, such as the decreased spheroid formation (Figure 4C,D). To date, targeting cancer stem-like cells yields only modest treatment effect in clinical trials, reflecting our lack of understanding of these unique but crucial subsets [46]. Though this result remains far from a clinical application, our identification of SOD2 as a crucial molecule in the induction of drug resistance helps to understand the role of the protein and the specific cells in the tumor tissue. This in turn has potential to improve cancer treatment with PI3K-inhibiting strategies in the future.

PI3K signals are known to be critical in the development of HNC [21], and the PI3K signaling pathway is widely believed to be a potential therapeutic target for treating HNC. Previous studies with mTOR inhibition, however, revealed feedback loops with the activation of upstream signaling [47]. This led to the development of the dual inhibitors that targeted PI3K/mTOR in the same cascades. However, results of the current study suggested eventual failure of the strategy. Multiple factors could be responsible for the acquired resistance, and here, we noticed the incomplete blocking of signals by activated AKT. This finding was in accordance with the report by Dufour et al. that suggested extra combinations with AKT inhibitors to achieve complete blockage of the pathway [48].

The tumor subsets with stemness features may have the advantage to survive from prolonged target therapy, thereby causing treatment failure. PI3K signals were literally known to be associated with the tumor-initiating features [49]; thus, it was not surprising to observe the enrichment of specific cell populations as a result of incomplete blockage of the signaling pathway. In fact, treatment with the PI3K/mTOR inhibitor resulted in higher proportion of ALDEFLUOR-positive cells (Figure 2A). This consequence from blocking the pathway is in accordance with reports using cells of other cancer types [31,50].

Tumor cell adaption in metabolic homeostasis contributes to promoting cell proliferation and the induction of drug resistance in cancer cells [51]. Accumulation of ROS could be controversial in determining cell fate. Emerging data suggested cell reaction to oxygen stress is associated with PI3K/mTOR inhibition and is often related to drug resistance. In chronic myeloid leukemia cells, AKT upregulation-induced ROS production attenuated the anticancer effects of imatinib by enriching progenitor cells, which in turn affected susceptibility of the target therapy [52]. In vascular smooth muscle cells, ROS was found to be responsible for the activation of AKT by angiotensin II [53]. In breast cancer cells, increased ROS levels in the drug resistant cells caused by retrieving from PI3K inhibitors during drug holidays would cause a proliferative defect in cells, suggesting a positive effect on tumor control [54]. Inhibition of the AKT pathway by a novel anti-mitotic agent, rigosertib, suggested induced ROS contributed to the treatment of HNC [55]. However, altered redox homeostasis caused by continuous interference from anti-cancer agents can enhance cell capacity in managing excessive ROS levels, and in turn, induct more resistant features [56].

Advancement in oncology suggests specific subsets of tumor-initiating/stem cells to promote resistance in cancer disease, including HNC [57]. Although it remains uncertain whether they are the decisive factors in the process, a clinical study already showed that higher expression of the stemness-featured biomarkers was associated with worse treatment outcome [58]. Anti-cancer drugs often result in an enriched stemness-featured subpopulation [59]. Enhancement of tumor-initiating cell features has been reported to accompany increased expression of SOD2, and as a result, strengthen the cells by superior ROS adjustment [45]. This can cause profound resistance, which in turn, affects treatment efficacy of the other mechanisms, such as cisplatin in resistant FaDu (Table 1) [44]. However, unlike target therapy, the resistance toward chemotherapy is often a net result of multiple complicated mechanisms [60], and thus, the cross-resistance was not similarly noted in the resistant UMSCC1 cells. In addition, considering that ALDEFLUOR cannot represent all tumor-initiating cells in HNC, the lack of SOD2 and resistance data in terms of the other selecting methods with CD44 or CD133 biomarkers is also a concern [61]. Nevertheless, results support an association of SOD2 with Bmi1, in accordance with the other reports [62,63]. Moreover, given that PI3K pathways are involved in regulation of ABC family of transport proteins such as ABCB1 and ABCG2 [50], and that PI3K/mTOR inhibitors were reported as substrates for these transporters, it was not surprising these cells could be involved in the acquired resistance [64]. Supportively in glioblastoma, the stemness featured subgroups with increased SOD2 that showed superior tolerance in oxidative stress, allowing them to take advantage in surviving cytotoxicity from another chemotherapy, temozolomide [45]. Altogether, these results support our finding of SOD2 as being a crucial factor that is associated with tumor-initiating cell features for enrichment of these HNC subsets, which in turn could contribute to multi-drug resistance.

In summary, acquisition of resistance may involve alterations of cell populations that have more predominant tumor-initiating cell features prone to survive harsh environments. These cells are capable for mitigating ROS toxicity that can contribute to the cross-resistance of other anti-tumor drugs. Importantly, inhibition of SOD2 leads to decreased spheroid formation, suggesting potential combination strategies in tumor treatment to overcome resistance. The study was made on cancer cell lines but not clinical samples, which was the major limitation. However, the application of ONCOMINE databases could suggest the findings have relevance for clinical disease. In addition, it is still unknown if the superior ROS clearance ability in this specific subgroup of cells has a direct correlation to enrichment of stemness features, and this question remains for future work.

## 4. Materials and Methods

### 4.1. Culture of the HNC Cell Lines and Derivation of the TMZ-Resistant Cells

The human HNC cell lines FaDu were purchased from American Type Culture Collection (Manassas, VA, USA) and UMSCC1 was a gift from Chia-Jui Yen (Department of Oncology, National Cheng Kung University, Tainan, Taiwan). DMEM with supplementation of 10% fetal bovine serum and penicillin/gentamicin antibiotics was used as medium (all from Invitrogen, Waltham, MA, USA). The resistant cells were derived from prolonged incubation with BEZ235 (Selleck, Houston, TX, USA) in the concentrations of 200 nM for FaDu and UMSCC1, and 1000 nM for FaDu. The compound in the beginning reduced the cellular proliferation and survival significantly, but eventually ended up with recovery in these parameters. Randomized single-cell clones were then selected and cultured. Both cells, FaDu and UMSCC1, were grown for at least 8 months (200r) and longer (1000r) to establish resistance. The resistant cells were maintained by 200 nM for 200r and 1000 nM for 1000r.

### 4.2. Western Blot Analysis

Whole cell lysate was separated in the SDS-PAGE by electrophoresis. It was then transferred onto polyvinylidene difluoride membranes (Bio-Rad, Hercules, CA, USA). The membranes were then blocked overnight in 5% nonfat milk with primary antibodies against SOD2 (1:3000, Cell Signaling, Danvers, MA, USA), Bmi1 (1:1000, GeneTex, Irvine, CA, USA), SOX2 (1:1000, GeneTex, Irvine, CA, USA), phosphorylated or total mTOR, phosphorylated or total mTOR, AKT (all from Cell Signaling, Danvers, MA, USA), and beta-actin (1:5000, Millipore, Burlington, MA, USA). After washing with phosphate buffered saline, they were incubated with secondary antibodies. The signals were then elicited with chemiluminescence substrate, and the intensity was detected with Amersham Hyperfilm ECL (GE Healthcare, Chicago, IL, USA).

### 4.3. Quantitative Real-Time Polymerase Chain Reaction (qPCR)

Total RNA was isolated from cells by standard procedure with TRIzol (Invitrogen, Waltham, MA, USA). It was then subjected to qRT-PCR with SuperScript II reagent (Invitrogen, Waltham, MA, USA). For preparation, the product was mixed with SYBR^®^ Green Master Mix (Applied Biosystems, Foster City, CA, USA) and the primers were the following: SOD2, F:5’-GGCCTACGTGAACAACCTGAA, R:5’-CTGTAACATCTCCCTTGGCCA; Nanog, F:5’-AATACCTCAGCCTCCAGCAGATG, R:5’-TGCGTCACACCATTGCTATTCTTC; Oct4, F:5′-CTTGCTGCAGAAGTGGGTGGAGGAA, R:5′-CTGCAGTGTGGGTTTCGGGCA; Bmi1, F:5’-TGGAGAAGGAATGGTCCACTTC, R:5’-GTGAGGAAACTGTGGATGAGGA; SOX2, F:5’-AAATGGGAGGGGTGCAAAAGAGGAG, R:5’-CAGCTGTCATTTGCTGTGGGTGATG; GAPDH, F:5′-GAAGGTGAAGGTCGGAGTC, R:5′-GAAGATGGTGATGGGATTC. The mixture was amplified and quantified by ABI 7000 Sequence Detection System (Applied Biosystems, Foster City, CA, USA). The data were then normalized to GAPDH using the 2^−ΔΔ*C*t^ formula.

### 4.4. Clonogenic Assay and Cell Density Assay

Cells were cultured in quantities of 400 or 5000–20,000 cells/mL/well in a 6- or 24-well plate with or without treatment for clonogenic assay or cell density assay, respectively. For clonogenic assay, the cells were washed after three days and incubated in treatment-free medium to allow them to grow into colonies. The number of colonies was then counted after staining and fixing of the cells with 50% ethanol containing 0.5% methylene blue (Sigma-Aldrich, St. Louis, MO, USA) for 90 min. For the cell density assay, cells were allowed proliferate with drug treatment. After three days the cells were stained and fixed with the aforementioned solution. For quantification, they were dissolved in 1% N-lauroyl-sarcosine (Sigma-Aldrich, St. Louis, MO, USA), then the optical density of the wavelength at 570 nm was measured by the microplate reader.

### 4.5. Detection of ALDH and ROS Expression by Flow Cytometry

Cells were pretreated and dissociated. For evaluation of the tumor-initiating feature, they were stained with ALDEFLUOR (STEMCELL Technologies, Vancouver, Canada) in presence or absence of the inhibitor diethylaminobenzaldehyde (DEAB; STEMCELL Technologies, Vancouver, Canada) to detect the specific marker ALDH [65]. In brief, cells were mixed in buffer-diluted ALDEFLUOR, and for control, was immediately transferred to DEAB. After incubation in 37 °C for 30 min, the cells were washed and resolved in the buffer. For ROS, they were labeled with dihydroethidium (Cayman, Ann Arbor, MI, USA). After incubation in 37 °C for 30 min, they were washed and resolved in PBS. The staining results of these cells were then studied and quantified for up to 10,000 cells using fluorescence-activated cell sorting (FACS) with a FACSCalibur system (BD Biosciences, Franklin Lakes, NJ, USA). The data were then analyzed via CellQuest (BD Biosciences, Franklin Lakes, NJ, USA).

The sorting process was through FACSAria™ III (BD Biosciences, Franklin Lakes, NJ, USA) for concomitant isolation of the ALDH-positive and the -negative cells, according to the ALDEFLUOR staining result. The cells were collected in the serum-free, growth factor-supplemented DMEM/F12 medium. At least 20,000 cells were used in each group for immediate RNA preparation, followed by the qPCR as described previously.

### 4.6. Tumor Spheroid Formation Assays

Cells were cultured in serum-free DMEM/F12 medium containing 2% B27 supplement, epidermal growth factor (20 ng/mL), and basic fibroblast growth factor (20 ng/mL) for supplementation. To promote anchorage-independent spheroid growth, methylcellulose (Sigma-Aldrich, St. Louis, MO, USA) with the concentration of 0.3% was added to the medium and was cultured in Costar® Ultra-Low Attachment Plates (Corning, Corning, NY, USA) [66]. For inhibition, a SOD inhibitor (SODi) sodium diethyldithiocarbamate trihydrate (Sigma-Aldrich, St. Louis, MO, USA) was applied for co-incubation.

### 4.7. Statistical Consideration

Statistical analysis of the data were obtained by using Prism 7 (version 7.03, GraphPad, La Jolla, CA, USA). The differences in continuous variables were calculated by Unpaired, Two-Tailed Student’s t-Test, with *p* value equaled 0.05 or less to be considered significant.

## Figures and Tables

**Figure 1 ijms-22-11260-f001:**
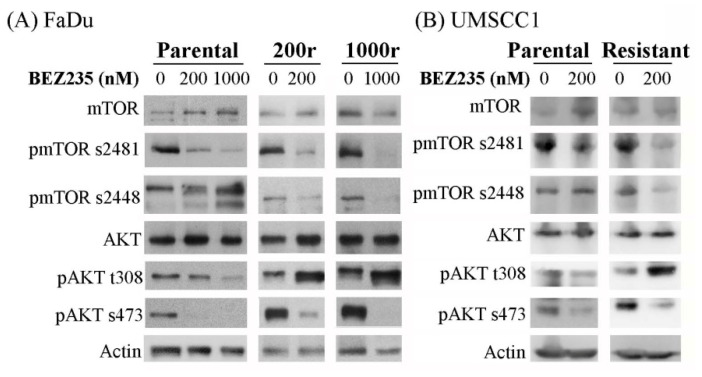
BEZ235 induces differential effects on the expression of the phosphorylated AKT (t308) in the drug sensitive FaDu/UMSCC1 cell lines and the FaDu/UMSCC1-dervided drug resistant cells (**A**), FaDu and the resistant cells; (**B**), UMSCC1 and the resistant cells). Expression of different proteins was examined by the Western blot analysis (abbreviation: p, phosphorylated).

**Figure 2 ijms-22-11260-f002:**
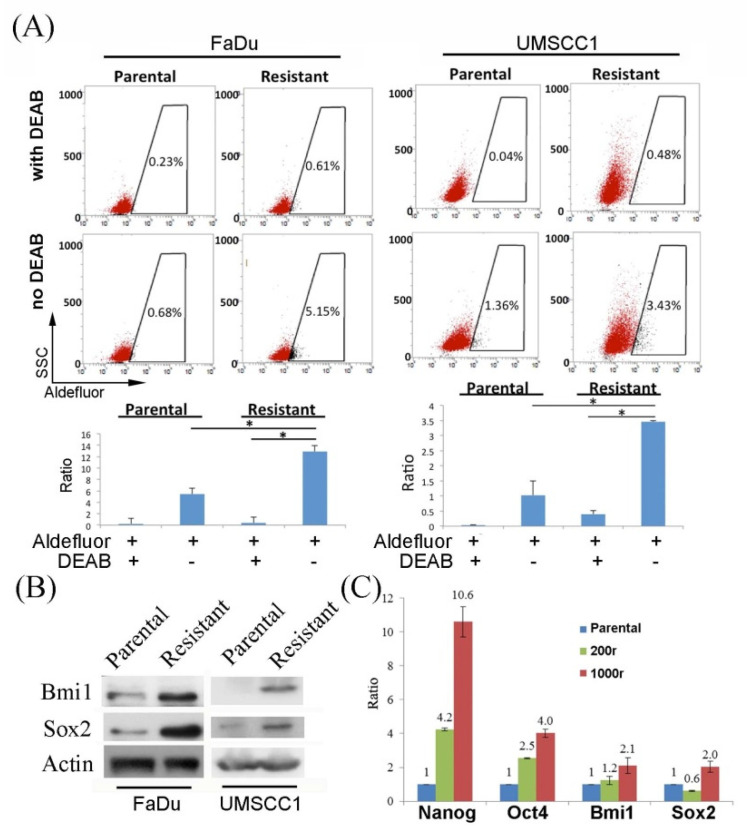
The multi-drug resistant cells exhibit features of the tumor-initiating cells. (**A**) Cells were stained with ALDEFLUOR for assessment of ALDH expression. The percentage of the gated area was annotated. (200r was used for the study of resistant FaDu; * *p* < 0.05.) (**B**) Western blot analysis of the expression of different stemness biomarkers in the parental and the drug-resistant cells was studied for the protein expression. (200r was used for the study of resistant FaDu). (**C**) Expression of the RNA transcripts of various stemness-related genes was examined by qPCR in the parental and the two FaDu resistant cells (200r and 1000r).

**Figure 3 ijms-22-11260-f003:**
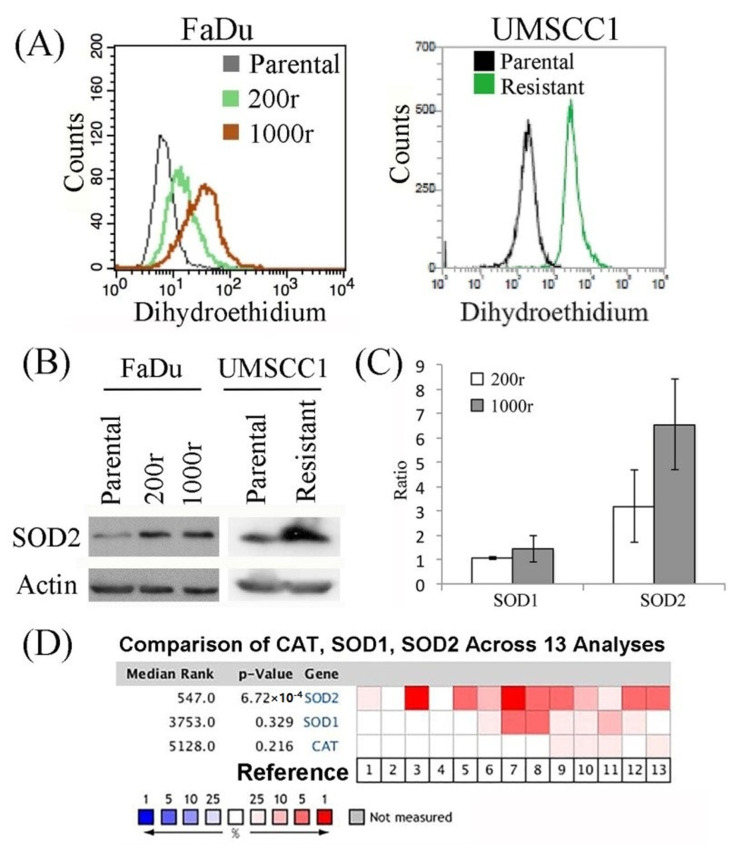
ROS analysis of the drug resistant cells. (**A**) The cells were assessed for the levels of ROS by staining with dihydroethidium, and the signals were measured with FACS. (**B**) The expression of SOD2 was examined by Western blot analysis. (**C**) The expression of the SOD1 and SOD2 mRNA transcript was measured by qPCR and calculated for the ratio to parental cells. (**D**) Gene expression of SOD1, SOD2, and catalase (CAT) in tumors versus normal tissue was obtained through ONCOMINE and are shown in the heatmap graph. The rank for a gene is the median rank for that gene across each of the analyses. The *p*-value for a gene is its *p*-value for the median-ranked analysis. [1. HNSCC (Cromer 2004); 2. Tongue SCC (Estilo 2009); 3. HNSCC (Ginos 2004); 4. Tongue SCC (Kuriakose 2004); 5. Oral cavity SCC (Peng 2011); 6. Floor of the mouth carcinoma; 7. Oral cavity carcinoma; 8. Oropharyngeal carcinoma; 9. Tongue carcinoma; 10. Tonsillar Ccarcinoma (Pyeon 2007); 11. Hypopharyngeal SCC (Schlingemann 2005); 12. Oral cavity SCC epithelia (Toruner 2004); 13. Tongue SCC (Ye 2008)].

**Figure 4 ijms-22-11260-f004:**
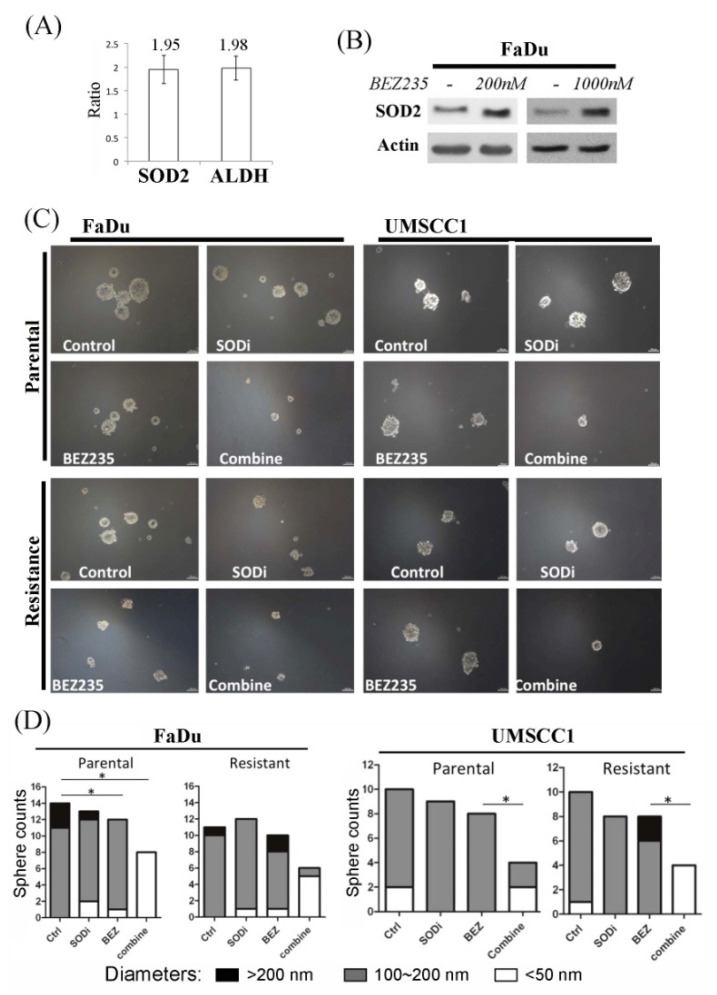
SOD2 expression is in association with stemness feature and BEZ235 treatment. (**A**) The ALDH-positive cells were sorted to compare with the ALDH-negative cells. The mRNA expression levels of SOD2 and ALDH were then analyzed by qPCR and standardized using ALDH-negative cells. (**B**) Tumor cells were treated with BEZ235 in different doses for 24 h. Cell lysates were extracted for protein analysis. (**C**) The parental and the resistant cells were incubated in serum-free culture medium for spheroid formation with or without BEZ235 and SOD inhibitor (SODi) (200r was used for the study of resistant FaDu). (**D**) Four photographs from the randomized fields of each condition (panel (**C**)) were taken under microscopy, and the numbers of spheroids were counted with the individual diameters recorded. The results were plotted in the bar graphs (* *p* < 0.05).

**Table 1 ijms-22-11260-t001:** IC_50_ of the drugs.

Drug	BEZ235	LY294002	Rapamycin	Gefitinib	Cisplatin
Category	PI3K/mTOR inhibitor	PI3K inhibitor	mTOR inhibitor	EGFR inhibitor	Platinum chemotherapy
FaDu *					
Parental	50 nM	19.8 μM	0.6 μM	1.0 μM	1.30 μM
200r	850 nM	44.7 μM	>10 μM	8.8 μM	3.85 μM
1000r	>1500 nM	43.45 μM	>10 μM	20 μM	4.59 μM
UMSCC1 *					
Parental	16.39 nM	18.14 μM	>50 μM	3.85 μM	5.64 μM
200r	>500 nM	44.35 μM	>50 μM	>50 μM	3.12 μM

* The resistant cells were developed by prolonged incubation with 200 nM (200r) or 1000 nM (1000r) BEZ235.

## Data Availability

Not applicable.

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
