# Peer review of "SOD2 Enhancement by Long-Term Inhibition of the PI3K Pathway Confers Multi-Drug Resistance and Enhanced Tumor-Initiating Features in Head and Neck Cancer"

_ijms, 2021, doi:10.3390/ijms222011260_

Round 1
Reviewer 1 Report
The authors created head and neck cancer cells that are resistant to the PI3K/mTOR inhibitor BEZ235. Resistant cells were characterized by tumor-infiltrating cells with persistent activation of phosphorylated AKT and increased expression of ADLH, Bmi1, Sox2 and SOD2. Co-incubation of SOD inhibitor and BEZ235 suggests a potential combination strategy to reduce spheroid cell development and overcome resistance.
It is known that the ABC family of transport proteins and ADLH are involved as the main mechanism by which cancer cells acquire the ability to resist antitumor drug resistance. Comments are needed on the possibility that ABCB1 and ABCG2 are involved in resistance to BEZ235.
The weakness of this paper is that it has not been investigated whether the addition of SOD inhibitors reduces the expression of ADLH, Bmi, and Sox2 in resistant cells as well as sphere formation. In addition, it is needed to examine the effect of the combination of SODi and BEZ235 on the proliferation of resistant cells.
The results of this study suggest that the mechanism by which resistance is acquired is due to persistent activation of phosphorylated AKT at threonine 308 and the over-expression of SOD2 in resistant HNSCC cells. Do AKT inhibitors affect drug resistance?
Table 1: The authors stated that resistant cells were also less sensitive to cisplatin. However, the IC50 of resistant UMSCC1 cells was lower than that of the parental cells. This needs to be explained.
Figure 2: ALDH-positive cells appears to account for a very small proportion. Do only ALDH-positive cells express high levels of SOD2?
The background of BEZ235 needs to be described in the introduction along with references.
Figure 3 legends: (B) and (C) needs to be exchanged.
Figure 3(A): what does "resistant" means? Is it 200r or1000r?
Figure 4: (D) is not explained in the figure legend and text. The authors need to explain the meaning of 50nM, 100-200nM, and 200nM. It is also necessary to explain why the efficiency of sphere formation is about the same in parental and resistant cells.
What is the SOD inhibitor. The authors need to include its name and where to get it origin.
Author Response
The authors created head and neck cancer cells that are resistant to the PI3K/mTOR inhibitor BEZ235. Resistant cells were characterized by tumor-infiltrating cells with persistent activation of phosphorylated AKT and increased expression of ADLH, Bmi1, Sox2 and SOD2. Co-incubation of SOD inhibitor and BEZ235 suggests a potential combination strategy to reduce spheroid cell development and overcome resistance.
Reply: Thank you for your comments, especially for suggestion of experiments to make our work more comprehensive. However, we noticed our study was less in mechanism research. This is because SOD2 has been learned associated with cancer stemness before, and because our focus on BEZ235 to enhance the gene and impact. As thus, we decided to revise our manuscript first to precisely intepret our findings and see if this would answer your concerns in the following point-by-point reply.
- It is known that the ABC family of transport proteins and ADLH are involved as the main mechanism by which cancer cells acquire the ability to resist antitumor drug resistance. Comments are needed on the possibility that ABCB1 and ABCG2 are involved in resistance to BEZ235.
Reply: Thank you for your suggestion. We modified the related content in the Discussion as the following. “Given that PI3K pathways has been involved in regulation of ABC family of transport proteins, e.g., ABCB1 and ABCG2 (Bleau AM 2009), and that PI3K/mTOR inhibitor was reported a substrate for these transporters (Wu CP 2020), it was not surprising this could be involved to the acquired resistance here.”
References:
Bleau AM, Hambardzumyan D, Ozawa T, Fomchenko EI, Huse JT, Brennan CW, Holland EC. PTEN/PI3K/Akt pathway regulates the side population phenotype and ABCG2 activity in glioma tumor stem-like cells. Cell Stem Cell. 2009 Mar 6;4(3):226-35. doi: 10.1016/j.stem.2009.01.007.
Wu CP, Huang YH,Lusvarghi S, Huang YH, Tseng PJ, Hung TH, Yu JS, Ambudkar SV. Overexpression of ABCB1 and ABCG2 contributes to reduced efficacy of the PI3K/mTOR inhibitor samotolisib (LY3023414) in cancer cell lines. Biochem Pharmacol. 2020 Oct;180:114137. doi: 10.1016/j.bcp.2020.114137. Epub 2020 Jul 4.
- The weakness of this paper is that it has not been investigated whether the addition of SOD inhibitors reduces the expression of ADLH, Bmi, and Sox2 in resistant cells as well as sphere formation. In addition, it is needed to examine the effect of the combination of SODi and BEZ235 on the proliferation of resistant cells.
Reply: We appreciate for the comment here. The reviewer pointed out the limitation of this study. Because the association and function of SOD2 in cancer stem cells have been studied in multiple tumor diseases, we did not go further in our study. Instead of repeating the experiments, we re-construct the associated content in the Discussion to minimize over-interpretation and to describe the current understanding regarding the roles of SOD2 in association with cancer stem cells and ROS as the following.
“The increased cellular level of SOD2 and the higher tolerance to ROS in the resistant cells implied its functional significance (Figures 3A, 3B, and 4A). SOD2 has been frequently reported mandatory for cancer stemness features (Kinugasa 2015, Chang 2014, Chien 2019). Supportively, inhibition of the protein led to decreased cells with one of the aforementioned features, spheroid formation, (Figure 4C).”
References:
Kinugasa H, Whelan KA, Tanaka K, et al. Mitochondrial SOD2 regulates epithelial-mesenchymal transition and cell populations defined by differential CD44 expression. Oncogene 2015;34:5229-39.
Chang CW, Chen YS, Chou SH, et al. Distinct subpopulations of head and neck cancer cells with different levels of intracellular reactive oxygen species exhibit diverse stemness, proliferation, and chemosensitivity. Cancer Res 2014;74:6291-305.
Chien CH, Chuang JY, Yang ST, et al. Enrichment of superoxide dismutase 2 in glioblastoma confers to acquisition of temozolomide resistance that is associated with tumor-initiating cell subsets. J Biomed Sci 2019;26:77.
- The results of this study suggest that the mechanism by which resistance is acquired is due to persistent activation of phosphorylated AKT at threonine 308 and the over-expression of SOD2 in resistant HNSCC cells. Do AKT inhibitors affect drug resistance?
Reply: Thank you for the comment. As shown in the results, AKT remained active with dual blockage capable inhibiting up signals and down signals of PI3K and mTOR, respectively. This can be a result of feedback, or compensatory regulation from alternative route such as MAPK pathway in activating mTORC1 (Faes S 2017). Based on this, direct inhibition against AKT attenuated the restoration of the pathway (Dufour 2013), warrenting the development of its specific inhibitor. As thus, we modified the discussion related to your comment as the following. “PI3K signals has known to be critical in HNC (Lui 2013), and was recently proved serviceable as the inhibitory targets. Experience from mTOR inhibition, however, revealed the feedback loops with the activation of upstream signaling (Rodrik-Outmezguine 2011). This led to development of the dual inhibitors that targeted PI3K/mTOR in the same cascades. Regardless of the initial response, however, the data from our resistant cell lines suggested eventual failure of the strategy. Multiple factors could be responsible for the acquired resistance, and hereby, we noticed restoration of the signals by activated AKT. This finding was in accordance with the report by Dufour, et al., in which suggested extra combination with AKT inhibitors to achieve complete blockage of the pathway (Dufour 2013).”
References::
Faes S, Demartines N, Dormond O. Resistance to mTORC1 Inhibitors in Cancer Therapy: From Kinase Mutations to Intratumoral Heterogeneity of Kinase Activity. Oxid Med Cell Longv 2017;2017:1726078. Doi: 10.1155/2017/1726078.
Lui VW, Hedberg ML, Li H, Vangara BS, Pendleton K, Zeng Y, Lu Y, Zhang Q, Du Y, Gilbert BR, Freilino M, Sauerwein S, Peyser ND, Xiao D, Diergaarde B, Wang L, Chiosea S, Seethala R, Johnson JT, Kim S, Duvvuri U, Ferris RL, Romkes M, Nukui T, Kwok-Shing Ng P, Garraway LA, Hammerman PS, Mills GB, Grandis JR. Frequent mutation of the PI3K pathway in head and neck cancer defines predictive biomarkers. Cancer Discov. 2013 Jul;3(7):761-9. doi: 10.1158/2159-8290.CD-13-0103.
Rodrik-Outmezguine VS, Chandarlapaty S, Pagano NC, Poulikakos PI, Scaltriti M, Moskatel E, Baselga J, Guichard, Rosen N. mTOR kinase inhibition causes feedback-dependent biphasic regulation of AKT signaling. Cancer Discov. 2011 Aug;1(3):248-59. doi: 10.1158/2159-8290.CD-11-0085.
Dufour M, Dormond-Meuwly A, Pythoud C, Demartines N, Dormond O. Reactivation of AKT signaling following treatment of cancer cells with PI3K inhibitors attenuates their antitumor effects. Biochem Biophys Res Commun. 2013 Aug 16;438(1):32-7. doi: 10.1016/j.bbrc.2013.07.014.
- Table 1: The authors stated that resistant cells were also less sensitive to cisplatin. However, the IC50 of resistant UMSCC1 cells was lower than that of the parental cells. This needs to be explained.
Reply: We noticed the inconsistant cisplatin resistance between cell types in Table 1. However, this could be resulted from complicated mechanism for chemotherapy resistance, which was unlike target therapy. In addition, diverse characteristics between cell lines to response to the drug could also have the roles in resulting the results. To avoid misunderstanding, we decided to amand the term “cross-resistance” in the Abstract and the Results to “profound resistance”, and have the “cross-resistance” issue discussed as the following.
“This should be awared because anti-cancer treatments, for example cisplatin, could have worse effect in cells with higher ROS tolerance (Chang 2014). However, unlike target therapy, the resistance toward chemotherapy is often net results of multiple complicated mechanisms (Chen 2019), and thus, the cross-resistance was not noted accordingly in the resisant UMSCC1 cells.”
Reference:
Chen SH, Chang JY. New Insights into Mechanisms of Cisplatin Resistance: From Tumor Cell to Microenvironment. Int J Mol Sci 2019;20:4136. Doi:10.3390/ijms20174136
Chang CW, Chen YS, Chou SH, et al. Distinct subpopulations of head and neck cancer cells with different levels of intracellular reactive oxygen species exhibit diverse stemness, proliferation, and chemosensitivity. Cancer Res 2014;74:6291-305.
- Figure 2: ALDH-positive cells appears to account for a very small proportion. Do only ALDH-positive cells express high levels of SOD2?
Reply: Thank you for your comment. Figure 4A showed ratio of SOD2 in ALDH-positive versus negative cells that were sorted by flow cytometry. This suggested 1.95-fold difference in SOD2 expression. However, this did not exclude any possibility of individual cell that had higher SOD2 expression, particularly in terms of ALDH can not represent all stemness-featured cells. In addition, we also revised legend of Figure 4A for better understanding of our application of
- The background of BEZ235 needs to be described in the introduction along with references.
Reply: We added the following content in the Introduction: “One of the attractive class of anticancer agents is the quinolone derivatives that targets both PI3K and mTOR simultaneously(Shiak A 2020). Of such, BEZ235 has demonstrated certain anticancer effects in a few cancer types. This dual inhibitor was reported to be more potent in cancer cells with PI3K aberrations such as the H1024R mutation, than to class I PI3K specific inhibitor (Wirtz 2015), exhibiting advantage of use since the aberrant PI3K pathway has commonly been shown to be related to the development of the disease (Liu 2013). While recent efforts had been made to improve the pharmacological properties of BEZ235(Tian 2019), we noticed its complexity in terms of the mechanism that can lead to chances for cells to develop cross-resistance as the consequence when the drug fails”
References:
Shaik A, Kirubakaran S. Evolution of PIKK family kinase inhibitors: A new age cancer therapeutics. Front Biosci (Landmark Ed). 2020 Mar 1;25:1510-1537. doi: 10.2741/4866.
Wirtz ED, Hoshino D, Maldonado AT, Tyson DR, Weaver AM. Response of head and neck squamous cell carcinoma cells carrying PIK3CA mutations to selected targeted therapies. JAMA Otolaryngol Head Neck Surg. 2015 Jun;141(6):543-9. doi: 10.1001/jamaoto.2015.0471.
Lui VW, Hedberg ML, Li H, Vangara BS, Pendleton K, Zeng Y, Lu Y, Zhang Q, Du Y, Gilbert BR, Freilino M, Sauerwein S, Peyser ND, Xiao D, Diergaarde B, Wang L, Chiosea S, Seethala R, Johnson JT, Kim S, Duvvuri U, Ferris RL, Romkes M, Nukui T, Kwok-Shing Ng P, Garraway LA, Hammerman PS, Mills GB, Grandis JR. Frequent mutation of the PI3K pathway in head and neck cancer defines predictive biomarkers. Cancer Discov. 2013 Jul;3(7):761-9. doi: 10.1158/2159-8290.CD-13-0103.
Tian L, Wang L, Qiao Y, Lu L, Lee P, Chang A, Ravi S, Rogers TA, Melancon MP. Antitumor Efficacy of Liposome-Encapsulated NVP-BEZ235 Combined with Irreversible Electroporation for Head and Neck Cancer. Molecules 2019 Oct 1;24(19):3560. doi: 10.3390/molecules24193560.
- Figure 3 legends: (B) and (C) needs to be exchanged.
Reply: Thank you for your correction. The legend of 3B and 3C has been exchanged in the revised version..
- Figure 3(A): what does "resistant" means? Is it 200r or1000r?
Reply: It was 200r. To clarify, we added a notification in the legend as the following:“(A) Cells were stained with Aldefluor for assessment of ALDH expression. The percentage of the gated area was annotated. (200r was used for the study of resistant FaDu)”
- Figure 4: (D) is not explained in the figure legend and text. The authors need to explain the meaning of 50nM, 100-200nM, and 200nM. It is also necessary to explain why the efficiency of sphere formation is about the same in parental and resistant cells.
Reply: Thank you for your comment. we revised the content in Figure 4D for some misleading labeling that was confusing for you, especially “nM” should be “nm” for measuring results of the sphere diameter.
Diversity between cell lines could cause this perplexed results, and thus, we had Figure 4D for information. As noted, parental FaDu was more sensitize to BEZ235 (P<0.05). This was not a case in parental UMSCC1, however. In terms of the resistant cells, we found reserved ability to form larger sphere in presence of BEZ235 (>200nm, bar in black color). This was reversed by combination of SODi. Notably, the size differed in FaDu resistant cells even though the decreased count did not reach significance, while UMSCC1 resistant cells reached statistical significance. We made the discussion in the Discussion part of revised content as the following: “Though the numbers of sphere was similarly affected by drugs in parental and resistant cells, we noticed latter prone to develop larger spheres particularly in presence of BEZ235 (Figure 4D). This aggressive feature was alleviated by combining with SODi, suggesting the crucial roles of the gene.”. In addition, we added legend of figure 4D as the following: “(D) Four photoraphs from the randomized fields of each condition (panel C) were taken under microscopy, and the numbers of spheroids were counted with the individual diameters recorded. The results were plotted in the bar graphs (*p < 0.05).”
- What is the SOD inhibitor. The authors need to include its name and where to get it origin.
Reply: It is sodium diethyldithiocarbamate trihydrate, which is a broad SOD inhibitor. To clarify, the following description is added to Materials and Methods. “For inhibition, a SOD inhibitor (SODi) sodium diethyldithiocarbamate trihydrate (Sigma-Aldrich) was applied for co-incubation”
Reviewer 2 Report
An interesting original article investigating the application of a dual inhibitor against Phosphoinositide-3-kinase and the mammalian target of rapamycin for the development of the resistant Head and neck cancer lines, showing high resistance to current therapies; I have some queries:
The English should be revised, as I found some sentences very hard to follow
You said you used an unpaired t-test to calculate significance, but it cannot be applied to all the tests you made; also, no p-value is present in the text; please check
page 2 line 45 you should add: traditional chemoterapic drugs used locally or sistemically do not seem to prove high effectiveness" and cite some articles such as: doi: 10.3390/curroncol28040213. and doi: 10.3390/medicina57060563
Thank You
Author Response
An interesting original article investigating the application of a dual inhibitor against Phosphoinositide-3-kinase and the mammalian target of rapamycin for the development of the resistant Head and neck cancer lines, showing high resistance to current therapies; I have some queries:
Reply: Thank you for your comment. Please find the following for point-by-point reply.
- The English should be revised, as I found some sentences very hard to follow
Reply: Thank you for your suggestion. We noticed our weak point in English. We have worked over the problem with an English native speaker, Dr. Chun Hei Antonio Cheung, in this revised manuscript. However, because of the very short due date, if you still feels further English editing is needed, we can send the manuscript to the professional editing once as per Journal’s Guide to Authors
- You said you used an unpaired t-test to calculate significance, but it cannot be applied to all the tests you made; also, no p-value is present in the text; please check
Reply: Thank you for your reminding. We used the statistical calulation for assessment in Figure 4D with the P level labeled usng asterisks to represent. For better understanding, we have added annotation in the figure legend of Figure 4D as following. “*p < 0.05”.
- page 2 line 45 you should add: traditional chemoterapic drugs used locally or sistemically do not seem to prove high effectiveness" and cite some articles such as: doi: 10.3390/curroncol28040213. and doi: 10.3390/medicina57060563
Reply: Thank you for the suggestion. We made modifcation in the Introduction as the following. “New discoveries in oncology showed certain aberrations in molecular signals that are critical for tumor to survive. Targeting drugs against these factors provide therapeutic opportunities that are more specific to tumor with less interference to the normal tissue. The specific strategies differ from conventional chemotherapies, which are often limited by toxicities to reach optimal treatment dose, or are forced to administrate via alternative route (Wen 2015, Bennardo 2021, Pentangelo 2021).”
References:
Wen Y, Grandis JR. Emerging drugs for head and neck cancer. Expert Opin Emerg Drugs. 2015 Jun;20(2):313-29. doi: 10.1517/14728214.2015.1031653.
Bennardo L, Bennardo F, Giudice A, Passante M, Dastoli S, Morrone P, Provenzano E, Patruno C, Nisticò SP. Local Chemotherapy as an Adjuvant Treatment in Unresectable Squamous Cell Carcinoma: What Do We Know So Far? Curr Oncol 2021 Jun 23;28(4):2317-2325. doi: 10.3390/curroncol28040213.
Pentangelo G, Nisticò SP, Provenzano E, Cisale GY, Bennardo L. Topical 5% Imiquimod Sequential to Surgery for HPV-Related Squamous Cell Carcinoma of the Lip. Medicina (Kaunas)2021 Jun 2;57(6):563. doi: 10.3390/medicina57060563.
Reviewer 3 Report
In the manuscript by Hsieh et al. the authors describe a study of Pi3K pathway inhibition resistance in HNC cell lines. While the manuscript is robust in employing many methods and 2 different cell lines, the data appears to be over-interpreted to some extent. The authors clearly show that generated resistant cell lines have higher drug IC50 levels, higher levels of some stem cell markers, higher ROS levels and higher SO2 levels. However, they do not actually test whether inhibiting SO2 changes any of the above findings. They switch to sphere forming assay as a substitute for all preceding tests while never establishing that such a substitution is justified.
Beyond the major problem above, there are significant language issues that need to be corrected, with at least 4 bigger problems on the first page alone (listed below). While the main message can be understood, the numerous errors make reading challenging and occasionally, the problems lead to ambiguities (ie. P2 L74).
P1 L34 – 37 Maybe better references should be provided. Comparing rates in 2018 and 2002-2012 might lead to wrong impressions.
P2 L92 Table 1 is split across pages. Table 1 presents the data only for 200nM BEZ235 while 2 cell lines created with different BEZ235 concentrations were used in other experiments.
P3 L103 Figure 1 panel B is unnecessarily smaller than panel A despite showing the same proteins
P4 L119 Figure 2. Panel A axis labels are unclear and/or missing. Panel B doesn’t indicate which resistant FaDu line was used (200r or 1000r) in the figure or its legend. This might be important since Sox2 levels do not agree between panels B and C. Panel C only has error bars for the 1000r columns? Were replicates performed and if so how? Gene order is different in panels B and C
P5 L142 Figure 3. Panels B and C and are inverted as their descriptions in the legend do not match the current figure. Color scale bar of Panel D is not explained. What is the meaning behind the p values and median ranks for the heatmap shown? Part of the legend appears to be highlighted in bold font.
P5 L159-161 The connection between the SOD2, stem cells and spheroids is not really evident.
P5 L161-163 this sentence is better suited to the discussion. However, the data shown still appear to be far from a clinical application or a treatment strategy?
P6L164 Figure 4 Panel C doesn’t specify which resistant FaDu cell line was used (200 or 1000r). Panel D legend is missing.
P6 L171 Discussion section. The data on mTOR and AKT phosphorylation is not discussed or put into context of current knowledge making this section of the results stand out as uninportant. yet it is prominent within the abstract.
P6 L173. The data do not show that. It was only shown that resistant cell lines have a subset of cells with high SO2 expression and that short term BEZ treatment increases SOD2 levels. What was not shown is that SO2 knockdown would decrease resistance. The spheroid experiment while interesting doesn’t really replace all previous experiments. At least IC50 experiment should be performed in the BEZ+SOD2 inhibitor context to really show that cells lose resistance when treated with SOD2i
P6 L178 while it is implied that SOD2 levels could confer higher resistance to ROS, this was not actually measured in the current study and thus the wording of this sentence is not justified by the data shown.
P7L236. Some details about the cell lines used should be provided. Especially so the results can be put in the context of Figure 3 panel D.
P7L239. How long were the cell lines exposed to BEZ235 to established resistance?
P8 L262 Primers for NANOG (Fig 2C) are not shown. On the other hand primers are shown for CD133 which is not shown in the results?
P8 L244. Spheroid formation section doesn’t follow the order of experiments shown in the results section.
P8 L287 the detection of ALDH and ROS section is lacking data on cell numbers, concentrations, sources of reagents, antibodies(?). Cell sorting is completely unclear.
Language and typos (page 1 only since they are too numerous to list comprehensively)
P1L16 “higher IC50 of the proliferation to BEZ235” language is problematic
P1L20 “increased expressed of ALDH level” language
P1L22 “within the resistance cells.” Should be resistant cells
P1L40-41 language
Author Response
- In the manuscript by Hsieh et al. the authors describe a study of Pi3K pathway inhibition resistance in HNC cell lines. While the manuscript is robust in employing many methods and 2 different cell lines, the data appears to be over-interpreted to some extent. The authors clearly show that generated resistant cell lines have higher drug IC50 levels, higher levels of some stem cell markers, higher ROS levels and higher SO2 levels. However, they do not actually test whether inhibiting SO2 changes any of the above findings. They switch to sphere forming assay as a substitute for all preceding tests while never establishing that such a substitution is justified.
Reply: We appreciate for the comment here. The reviewer pointed out the limitation of this study, and thus, we agreed some of our discussion appeared to be over-interpreted. The functional analysis of SOD2 in cancer stem cells and ROS has been studied in multiple tumor diseases from our (Chien 2019) and the other groups (Chang 2014). Instead of repeating the experiments, we re-construct the associated content in the Discussion to minimize over-interpretation and to describe the current understanding regarding the roles of SOD2 in association with cancer stem cells and ROS as the following.
“Anti-cancer treatment often accompany with enhancement of the stemness-featured subpopulation (Liu 2013). Enhancement of tumor-initiating cell features has been reported accompanied with increased SOD2 expression with superior ROS adjustment (Chien 2017, Chang 2014). The intractable features then causes profound resistance, and with more concern, is toward cross-resistance to treatment of the other mechanisms, i.e., less sensitive to cisplatin in the resistant FaDu (Table 1). This should be awared because anti-cancer treatments, for example cisplatin, could have worse effect in cells with higher ROS tolerance (Chang 2014).”
References:
Liu YP, Yang CJ, Huang MS, et al. Cisplatin selects for multidrug-resistant CD133+ cells in lung adenocarcinoma by activating Notch signaling. Cancer Res 2013;73:406-16.
Chang CW, Chen YS, Chou SH, et al. Distinct subpopulations of head and neck cancer cells with different levels of intracellular reactive oxygen species exhibit diverse stemness, proliferation, and chemosensitivity. Cancer Res 2014;74:6291-305.
Chien CH, Chuang JY, Yang ST, et al. Enrichment of superoxide dismutase 2 in glioblastoma confers to acquisition of temozolomide resistance that is associated with tumor-initiating cell subsets. J Biomed Sci 2019;26:77.
- Beyond the major problem above, there are significant language issues that need to be corrected, with at least 4 bigger problems on the first page alone (listed below). While the main message can be understood, the numerous errors make reading challenging and occasionally, the problems lead to ambiguities (ie. P2 L74).
Reply: Thank you for your suggestion. We noticed our weak point in English. We have worked over the problem with an English native speaker, Dr. Chun Hei Antonio Cheung, in this revised manuscript. However, because of the very short due date, if you still feels further English editing is needed, we can send the manuscript to the professional editing once as per Journal’s Guide to Authors
- P1 L34 – 37 Maybe better references should be provided. Comparing rates in 2018 and 2002-2012 might lead to wrong impressions.
Reply: To make a better comparison for 2002-2012, as well as to reflect the latest data, we modified the related content in the Introduction as the following. “The worldwide incidence of the disease rapidly grows from 2008 of 263,900 to 2018 of 890,000 cases, causing 128,000 and 450,000 deaths, respectively, and ranking sixth of the most common cancer nowadays in the world (Jemal 2011; Johnson 2020)”
References:
Jemal A, Bray F, Center MM, Ferlay J, Ward E, Forman D: Global cancer statistics. CA Cancer J Clin 2011, 61:69–90.
Johnson DE, Burtness B, Leemans CR, Lui VWY, Bauman JE, Grandis JR. Head and neck squamous cell carcinoma. Nat Rev Disease Primers 2020;6:92.
- P2 L92 Table 1 is split across pages. Table 1 presents the data only for 200nM BEZ235 while 2 cell lines created with different BEZ235 concentrations were used in other experiments.
Reply: Thank you for your reminder. The IC50 results of FaDu-1000r cell line are added to the table.
- P3 L103 Figure 1 panel B is unnecessarily smaller than panel A despite showing the same proteins
Reply: Thank you for your suggestion. Figures 1A and 1B are now re-edited into the same resolution.
- P4 L119 Figure 2. Panel A axis labels are unclear and/or missing. Panel B doesn’t indicate which resistant FaDu line was used (200r or 1000r) in the figure or its legend. This might be important since Sox2 levels do not agree between panels B and C. Panel C only has error bars for the 1000r columns? Were replicates performed and if so how? Gene order is different in panels B and C
Thank you for your comment. We add the label of the axis in figure 2A as your reminder. For Figure 2B, 200r was used as the resistant FaDu line. The relevant description is added in the legend: “Western Blot analysis of the expression of different stemness biomarkers in the parental and the drug-resistant cells was studied for the protein expression. (200r was used for the study of resistant FaDu).” For panel C, the error bars was missing, which in turns made the data not proper to intepret. We would like to thank you for the reminder. The error bar is added and the gene order is changed in the revised version to represent the triplicated data.
“The mRNA study by real-time PCR revealed concomitant result of increased Bmi1 (figure 2C). Despite of unexpectedly decreased level of Sox2 gene in FaDu-r200 cells, it was still enhanced in FaDu-r1000. In addition, we also found increased expression of Nanog and Oct4 genes in all resistant cells for support.”
- P5 L142 Figure 3. Panels B and C and are inverted as their descriptions in the legend do not match the current figure. Color scale bar of Panel D is not explained. What is the meaning behind the p values and median ranks for the heatmap shown? Part of the legend appears to be highlighted in bold font.
Reply: For the methods applied in Oncomine, Rhodes DR and colleagues had a detail explaination (ONCOMINE: A Cancer Microarray Database and Integrated Data-Mining Platform. Neoplasia 2004)
“Following the assignment of samples to classes, each gene was assessed for differential expression with t-statistics using Total Access Statistics 2002 (FMS Inc., Vienna, VA). t-Tests were conducted both as two-sided for differential expression analysis and one-sided for specific overexpression analysis.”
Our study utilized the platform for gene analysis between datasets. The data in Figure 3D showed the unmodified heatmap regarding SOD1, SOD2, and CAT genes. Here, we resume the description of t-test and p-value from the platform as the following. “The rank for a gene is the median rank for that gene across each of the analyses. The p-Value for a gene is its p-Value for the median-ranked analysis.”
- P5 L159-161 The connection between the SOD2, stem cells and spheroids is not really evident.
Reply: Thank you for your comment. We agree the connection is weak and the use of speroid formation assay is only represent partial feature of stemness. Hence, we revised the description here to better fit the results in Figure 4 as the following.
“Because previous reports have shown associations between SOD2 and the cancer stemnes features (Kinugasa 2015, Chang 2014, Chien 2019), we postulated the treatment could have adverse effect in bringing cell tolerance, and hence, selecting subgroups of stem-like cells that impede further treatment. To mitigate the effect, we found that co-incubation of SOD inhibitors with BEZ235 could significantly eliminate development of spheroid cells (figure 4C). By this mean, our study suggested a potential treatment strategy to minimize accumulation of this specific aggressive subets by combination therapy, and avoidance of profound resistance.”
References:
Kinugasa H, Whelan KA, Tanaka K, et al. Mitochondrial SOD2 regulates epithelial-mesenchymal transition and cell populations defined by differential CD44 expression. Oncogene 2015;34:5229-39.
Chang CW, Chen YS, Chou SH, et al. Distinct subpopulations of head and neck cancer cells with different levels of intracellular reactive oxygen species exhibit diverse stemness, proliferation, and chemosensitivity. Cancer Res 2014;74:6291-305.
Chien CH, Chuang JY, Yang ST, et al. Enrichment of superoxide dismutase 2 in glioblastoma confers to acquisition of temozolomide resistance that is associated with tumor-initiating cell subsets. J Biomed Sci 2019;26:77.
- P5 L161-163 this sentence is better suited to the discussion. However, the data shown still appear to be far from a clinical application or a treatment strategy?
Reply: Thank you for the suggestion. We will move the sentence to the Discussion. In addition, we will discuss the clinical application problem as the following: “To date, targeting cancer stem-like cells yields only modest treatment effect in clinical trials, reflecting our less understanding of these unique but crucial subsets (Zhou 2021).Though remains far from a clinical application, our identification of SOD2 as a crucial molecule in the induction of drug resistance helps to understand the role of the protein and the specific cells in the tumor tissue. This in turn, has potential to help improving the cancer treatment with PI3K inhibiting strategies in the future.”
Reference:
Zhou HM, Zhang JG, Zhang X, Li Q. Targeting cancer stem cells for reversing therapy resistance: mechanism, signaling, and prospective agents. Signal Transduct Target Ther. 2021 Feb 15;6(1):62. doi: 10.1038/s41392-020-00430-1.
- P6L164 Figure 4 Panel C doesn’t specify which resistant FaDu cell line was used (200 or 1000r). Panel D legend is missing.
Reply: We added following content in panel C legend: “(C)The parental and the resistant cells were incubated in serum-free culture medium for spheroid formation with or without BEZ235 and SOD inhibitor (SODi). (200r was used for the study of resistant FaDu)”. We also added legend for panel D as the following: “(D) Four photoraphs from the randomized fields of each condition (panel C) were taken under microscopy, and the numbers of spheroids were counted with the individual diameters recorded. The results were plotted in the bar graphs (*p < 0.05).”
- P6 L171 Discussion section. The data on mTOR and AKT phosphorylation is not discussed or put into context of current knowledge making this section of the results stand out as uninportant. yet it is prominent within the abstract.
Reply: Thank you for your valuable comment. We noticed our lacking in discussion regarding the data on mTOR and AKT, and thus, modified the discussion related to your comment as the following. “PI3K signals are known to be critical in the development of HNC(Lui 2013), and the PI3K signaling pathway has widely believed as a potential therapeutic target for treating HNC. Previous studies with mTOR inhibition, however, revealed the feedback loops with the activation of upstream signaling (Rodrik-Outmezguine 2011). This led to the development of the dual inhibitors that targeted PI3K/mTOR in the same cascades. However, results of the current study suggested eventual failure of the strategy. Multiple factors could be responsible for the acquired resistance, and here, we noticed the incomplete blocking of signals by activated AKT. This finding was in accordance with the report by Dufour, et al., in which suggested extra combination with AKT inhibitors to achieve complete blockage of the pathway(Dufour 2013).”
References::
Lui VW, Hedberg ML, Li H, Vangara BS, Pendleton K, Zeng Y, Lu Y, Zhang Q, Du Y, Gilbert BR, Freilino M, Sauerwein S, Peyser ND, Xiao D, Diergaarde B, Wang L, Chiosea S, Seethala R, Johnson JT, Kim S, Duvvuri U, Ferris RL, Romkes M, Nukui T, Kwok-Shing Ng P, Garraway LA, Hammerman PS, Mills GB, Grandis JR. Frequent mutation of the PI3K pathway in head and neck cancer defines predictive biomarkers. Cancer Discov. 2013 Jul;3(7):761-9. doi: 10.1158/2159-8290.CD-13-0103.
Rodrik-Outmezguine VS, Chandarlapaty S, Pagano NC, Poulikakos PI, Scaltriti M, Moskatel E, Baselga J, Guichard, Rosen N. mTOR kinase inhibition causes feedback-dependent biphasic regulation of AKT signaling. Cancer Discov. 2011 Aug;1(3):248-59. doi: 10.1158/2159-8290.CD-11-0085.
Dufour M, Dormond-Meuwly A, Pythoud C, Demartines N, Dormond O. Reactivation of AKT signaling following treatment of cancer cells with PI3K inhibitors attenuates their antitumor effects. Biochem Biophys Res Commun. 2013 Aug 16;438(1):32-7. doi: 10.1016/j.bbrc.2013.07.014.
- P6 L173. The data do not show that. It was only shown that resistant cell lines have a subset of cells with high SO2 expression and that short term BEZ treatment increases SOD2 levels. What was not shown is that SO2 knockdown would decrease resistance. The spheroid experiment while interesting doesn’t really replace all previous experiments. At least IC50 experiment should be performed in the BEZ+SOD2 inhibitor context to really show that cells lose resistance when treated with SOD2i
Reply: Thank you for your comment. Regarding the sentence you mentioned, the original content is seemingly over interpreted. As thus, we decided to modify the discussion as the following based on your suggestion. “Here, we showed that the drug resistant cells have higher SOD2 expression level, and PI3K/mTOR inhibition increased the SOD2 level in cells. Failure to inhibit against the pathway had profound resistance against anti-tumor treatments which have similar mechanism as well as those with more different mechanism (Figure 1)”
- P6 L178 while it is implied that SOD2 levels could confer higher resistance to ROS, this was not actually measured in the current study and thus the wording of this sentence is not justified by the data shown.
Reply: Thank you for your comment. Regarding the sentence you mentioned, the original content is seemingly with over interpretation. As thus, we decided to modify the discussion as the following. “The increased cellular level of SOD2 and the higher tolerance to ROS in the resistant cells implied its functional significance (Figures 3A, 3B, and 4A). SOD2 has been frequently reported mandatory for cancer stemness features [38,39,57]. Supportively, inhibition of the protein led to decreased cells with one of the aforementioned features, spheroid formation, (Figure 4C).”
- Some details about the cell lines used should be provided. Especially so the results can be put in the context of Figure 3 panel D.
Reply: Thank you for the concern of study limitation. As to respond to your suggestion, we revised the conclusion as the following: “The study was made on cancer cell lines but not clinical samples, which was the major limitation. However, the application of Oncomine databases could suggest the findings in relevance with clinical disease. In addition, it is still not known whether superior ROS clearance in this specific subgroup of cells has direct correlation to enrichment of stemness features, and remain to be elucidated in the future.”
- How long were the cell lines exposed to BEZ235 to established resistance?
Reply: Both cells, FaDu and UMSCC1 were incubated with BEZ235 persistantly for at least 8 months (200r) and longer (1000r), including the procedure of clone selection.
- P8 L262 Primers for NANOG (Fig 2C) are not shown. On the other hand primers are shown for CD133 which is not shown in the results?
Reply: Thank you for your reminder that allows us to correct. The CD133 primer was an error here. We correct the content with Nanog sequence, . We also add Oct4 sequence, which is used for detection in Figure 2C.
- P8 L244. Spheroid formation section doesn’t follow the order of experiments shown in the results section.
Reply: Thank you for your reminder. We move the “Tumor spheroid formation assay” section to the last second section of the Materials and Methods.
- P8 L287 the detection of ALDH and ROS section is lacking data on cell numbers, concentrations, sources of reagents, antibodies(?). Cell sorting is completely unclear.
Reply: For your information, 10,000 cells were analyzed for each single test in FACS, and the preparation followed manufacturers’ protocal with minimal modifcation. To clearly describe the procedure, we revised the related content in Materials and Methods as the following. “Cells were pretreated and dissociated. For evaluation of the tumor-initiating feature, they were stained with ALDEFLUOR (STEMCELL Technologies, Vancouver, Canada) in presence or absence of the inhibitor diethylaminobenzaldehyde (DEAB; STEMCELL Technologies) to detect the specific marker ALDH [25]. In brief, cells were mixed in buffur-diluted ALDEFLUOR, and for control, was immediately transferred to DEAB. After incubation in in 37 degree Celcius for 30 minutes, the cells were washed and resolved in the buffer. For ROS, the cells were labeled with PBS-diluted dihydroethidium (Cayman, Ann Arbor, MI, USA). After incubation in 37 degree Celcius for 30 minutes, they were washed and resolved in PBS. The staining results of these cells were then studied and quantified for up to 10,000 cells using fluorescence-activated cell sorting (FACS) with FACSCalibur system (BD Biosciences, Franklin Lakes, NJ, USA). The data was then analyed via CellQuest (BD Biosciences).”
- Language and typos (page 1 only since they are too numerous to list comprehensively)
Reply: Please note the following points are corrected and revised with English editing.
P1L16 “higher IC50 of the proliferation to BEZ235” language is problematic
P1L20 “increased expressed of ALDH level” language
P1L22 “within the resistance cells.” Should be resistant cells
P1L40-41 language
Round 2
Reviewer 3 Report
The revised manuscript by Hsieh et al greatly improved the language as well as addressed many of the issues found with the original. The manuscript is now a much improved version with better flow of the text and better connections of separate experiments. Unfortunately, the clean version (without track changes) was not present which made the review a bit challenging and possibly some remaining typos are simply due to revisions (deletions) remaining in the PDF.
In the authors reply, the authors state that the cell lines were grown for 8 or more months to establish resistance. For the sake of completeness (and that the other authors could more easily replicate the methods) this information should be provided in the manuscript as well, for example at P13 L379.
While not an expert in flow cytometry, the section 4.5 (P15 L431, flow cytometry) possibly includes ambiguities. FACSCalibur was used for regular flow-cytometry and not sorting as implied by the text. On the other hand, FACSAria was used for actual sorting of ALDH positive cells, but the method is still poorly described to the point that another researcher cannot replicate the study. How many cells were actually sorted out as ALDH positive? Were they somehow further expanded or was the small number adequate for mRNA analysis? How many ALDH negative cells were collected and further processed for qPCR?
Typos and minor (possibly not comprehensive due to time limitations)
P2 L66 references [7,8,9] can be replaced by [7-9]
P2L73 “are highly relied” should be highly reliant
P2L81 “imped” should be impede
P4 L141 double commas remain after revisions “BEZ235,, “
P5 L167 after revision the sentence is not correct “…Alderfluor, is one, of the tumor-initiating cells.”
P7 L207 the sentence after revision still has some problems“Hereby, the drug resistant cellsderivations showed enhanced protein protein expression of SOD2,…”
P9 L234 potential revision problem “Inhibition of SOD2 resulted in less expression of reduced the stemness features”
Author Response
- The revised manuscript by Hsieh et al greatly improved the language as well as addressed many of the issues found with the original. The manuscript is now a much improved version with better flow of the text and better connections of separate experiments. Unfortunately, the clean version (without track changes) was not present which made the review a bit challenging and possibly some remaining typos are simply due to revisions (deletions) remaining in the PDF.
Reply: Thank you for your recognition over our effort in improving the language quality. We beg your pardon for not preparing the clean version. The clean version will be attached for your reference along with this revision.
In addition, for your reference, please also note the correction of the first author's spelling in the manuscript from Hsieh to Hsueh.
- In the authors reply, the authors state that the cell lines were grown for 8 or more months to establish resistance. For the sake of completeness (and that the other authors could more easily replicate the methods) this information should be provided in the manuscript as well, for example at P13 L379.
Reply: Thank you for your suggestion. To clarify, we revised section 4.1., Culture of the HNC cell lines and derivation of the TMZ-resistant cells, with the following content: “Both cells, FaDu and UMSCC1, were grown for at least 8 months (200r) and longer (1000r) to establish resistance.”
- While not an expert in flow cytometry, the section 4.5 (P15 L431, flow cytometry) possibly includes ambiguities. FACSCalibur was used for regular flow-cytometry and not sorting as implied by the text. On the other hand, FACSAria was used for actual sorting of ALDH positive cells, but the method is still poorly described to the point that another researcher cannot replicate the study. How many cells were actually sorted out as ALDH positive? Were they somehow further expanded or was the small number adequate for mRNA analysis? How many ALDH negative cells were collected and further processed for qPCR?
Reply: Our sorting procedure allowed us to collect at least 100,000 ALDH-positive cells, simultaneously accompanying with about 6-fold of that numbers for the ALDH-negative cells. All cells were temporarily collected into the serum-free tumor spheroid-culturing medium to reserve their stemness features, if any. Later, roughly 20,000 to 50,000 cells in ALDH-positive cells, or more in ALDH-negative cells, were used for immediate preparation of RNA followed by qPCR analysis. This ensured the number of cells were adequate for gene study.
To clarify, we revised section 4.5., Detection of ALDH and ROS expression by flow cytometry, with the following content: “For sorting, the process was through FACSAria™ III (BD Biosciences) for concomitant isolation of the ALDH-positive and -negative cells according to the ALDEFLUOR staining result. The cells were collected into the serum-free, growth factors-supplemented DMEM/F12 medium. At least 20,000 cells in each group were used for immediate RNA preparation followed by the qPCR as previously described.”
- Typos and minor (possibly not comprehensive due to time limitations)
Reply: Thank you for your comment in the typos and the language errors. These, as well as some others found by us, are corrected or modified in the revised manuscript. In addition, some sentences were reconstructed with least change over the meaning and the reference to improve the readability, with majority in the last second paragraph of the Discussion. Please find the Track Changes for the detail information.
- P2 L66 references [7,8,9] can be replaced by [7-9]
Reply: Thank you for the comment. This has been corrected.
- P2L73 “are highly relied” should be highly reliant
Reply: Thank you for the comment. This has been corrected.
- P2L81 “imped” should be impede
Reply: Thank you for the comment. This has been corrected.
- P4 L141 double commas remain after revisions “BEZ235,, “
Reply: Thank you for the comment. This has been corrected.
- P5 L167 after revision the sentence is not correct “…Alderfluor, is one, of the tumor-initiating cells.”
Reply: Thank you for your comment. We revised it as the following: “…ALDEFLUOR, is one of the markers for tumor-initiating cells.”
- P7 L207 the sentence after revision still has some problems“Hereby, the drug resistant cellsderivations showed enhanced protein protein expression of SOD2,…”
Reply: Thank you for your comment. We revised it as the following: “Here, the drug-resistant cells showed enhanced protein expression of SOD2 (figure 3B), as compared to the parental drug sensitive cells.”
- P9 L234 potential revision problem “Inhibition of SOD2 resulted in less expression of reduced the stemness features”
Reply: Thank you for your comment. We revised it as the following: “Inhibition of SOD2 resulted in reduced stemness features.”